# Mutations That Affect the Surface Expression of TRPV6 Are Associated with the Upregulation of Serine Proteases in the Placenta of an Infant

**DOI:** 10.3390/ijms222312694

**Published:** 2021-11-24

**Authors:** Claudia Fecher-Trost, Karin Wolske, Christine Wesely, Heidi Löhr, Daniel S. Klawitter, Petra Weissgerber, Elise Gradhand, Christine P. Burren, Anna E. Mason, Manuel Winter, Ulrich Wissenbach

**Affiliations:** 1Experimental and Clinical Pharmacology and Toxicology, Center for Molecular Signaling (PZMS), Saarland University, Buildings 61.4 and 46, 66421 Homburg, Germany; Claudia.Fecher-Trost@uks.eu (C.F.-T.); Karin.Wolske@uks.eu (K.W.); Christine.Wesely@uks.eu (C.W.); Heidi.Loehr@uks.eu (H.L.); s8daklaw@teams.uni-saarland.de (D.S.K.); Petra.Weissgerber@uks.eu (P.W.); Manuel.Winter@uks.eu (M.W.); 2Transgenic Technologies, Center for Molecular Signaling (PZMS), Saarland University, Building 61.4, 66421 Homburg, Germany; 3Kinder- und Perinatalpathologie Dr. Senckenberg, Institut für Pathologie Universitätsklinikum Frankfurt/Main Theodor-Stern-Kai 7, 60590 Frankfurt, Germany; Elise.Gradhand@kgu.de; 4Department of Translational Health Sciences, Bristol Medical School, University of Bristol, University Hospitals Bristol and Weston NHS Foundation Trust, Upper Maudlin St, Bristol BS2 8BJ, UK; Christine.Burren@uhbw.nhs.uk; 5Histopathology Department, Aneurin Bevan University Health Board, Royal Gwent Hospital, Cardiff NP20 2UB, UK; Anna.Mason@wales.nhs.uk

**Keywords:** TRPV6, placenta, calcium transport, skeletal dysplasia, serine proteases, subunit assembly, transient receptor potential

## Abstract

Recently, we reported a case of an infant with neonatal severe under-mineralizing skeletal dysplasia caused by mutations within both alleles of the *TRPV6* gene. One mutation results in an in frame stop codon (R_510_stop) that leads to a truncated, nonfunctional TRPV6 channel, and the second in a point mutation (G_660_R) that, surprisingly, does not affect the Ca^2+^ permeability of TRPV6. We mimicked the subunit composition of the unaffected heterozygous parent and child by coexpressing the TRPV6 G_660_R and R_510_stop mutants and combinations with wild type TRPV6. We show that both the G_660_R and R_510_stop mutant subunits are expressed and result in decreased calcium uptake, which is the result of the reduced abundancy of functional TRPV6 channels within the plasma membrane. We compared the proteomic profiles of a healthy placenta with that of the diseased infant and detected, exclusively in the latter two proteases, HTRA1 and cathepsin G. Our results implicate that the combination of the two mutant TRPV6 subunits, which are expressed in the placenta of the diseased child, is responsible for the decreased calcium uptake, which could explain the skeletal dysplasia. In addition, placental calcium deficiency also appears to be associated with an increase in the expression of proteases.

## 1. Introduction

TRPV6 is a Ca^2+^ selective ion channel which shows a very restricted expression pattern. Human TRPV6 is expressed in a few glands, including acinar salivary and lacrimal glands, in parts of the small intestine, and in the trophoblast layer of the placenta [1,2,3]. In addition, TRPV6 is overexpressed in a number of malignancies, namely, prostate, mammary ovarial and endometrial cancer [1,2,4,5,6,7,8,9,10]. In the human population, two TRPV6 alleles, TRPV6a and TRPV6b, exist, leading to a coupled polymorphism with three distinct amino acid exchanges detected in TRPV6a (R_197_V_418_T_721_) and TRPV6b (C_197_M_418_M_721_) [1,11,12]. Whether this polymorphism has a functional consequence is not known. In recent publications, the effects of TRPV6 mutations altering the functionality of TRPV6 channels in humans were published [13,14,15,16,17]. Dysfunction of TRPV6 channels leads to transient neonatal hyperparathyroidism (HRPTTN) and is listed in the OMIM database (Online Mendelian Inheritance in Man). We recently described the case of an infant who suffers from neonatal severe under mineralizing skeletal dysplasia due to underlying severe transient hyperparathyroidism. Both the *TRPV6* alleles of the infant showed mutations (13): one mutation leads to an amino acid exchange of glycine 660 to arginine (G_660_R) at the C-terminus of the TRPV6 protein, which is presumed to be localized intracellularly. The second *TRPV6* locus exhibits a mutation which leads to an in frame stop codon replacing an arginine coding triplet by a stop codon, R_510_stop (stop mutant). The TRPV6 protein contains six hydrophobic transmembrane domains, and the pore region of the channel is located between the fifth and the sixth domain [18]. The R_510_stop mutation is placed in the linker sequence between the fourth and the fifth transmembrane domain and results in a truncated protein without a pore region and any detectable Ca^2+^ permeability. We demonstrate that the mutations of the affected child lead to an inadequate channel assembly and, as a consequence, to a reduced insertion of the maternal G_660_R-mutant in combination with the truncated paternal TRPV6-R_510_stop mutant into the plasma membrane. 

In addition, we show by mass spectrometry that two serine proteases were only detectable in the placenta of the affected child. In addition, a protease is upregulated in a TRPV6 expressing human trophoblast cell line cultured under a low Ca^2+^ condition.

## 2. Results

### 2.1. Functional Consequence of Mutations within TRPV6 Channel Subunits

An affected child who exhibits mutations within the *TRPV6* gene was recently analysed using whole exome sequencing [13,14]. The child showed a pronounced dysplasia of the skeleton and died after several months. One *TRPV6* allele of the child contained a mutation that leads to a G_660_R mutation in the very C-terminus of the coding sequence, whereas the second allele contained an in frame stop codon, R_510_stop, which leads to a truncated protein without the pore region of the TRPV6 channel. We focused on the TRPV6 mutations and cloned a number of TRPV6 constructs in the dicistronic pCAGGS-IRES-GFP or IRES-RFP vectors, allowing the expression of TRPV6 independently from the fluorescent proteins. First, we analysed the G_660_R mutation present in the affected child. We introduced this mutation in the TRPV6 cDNA and expressed the construct in HEK293 cells, measured Ca^2+^ uptake, and compared the result with wild type TRPV6 expressing cells (Figure 1A,B). Surprisingly, the Ca^2+^ uptake is not significantly different compared to wild type TRPV6 expressing cells (Figure 1B). The peak value of the two constructs was not altered. TRPV6 channels consist of four identical subunits and, in the human placenta, both TRPV6 loci are expressed [1,19,20,21]. Therefore, we mimicked the TRPV6 expression of the nonaffected parents and the affected child by coexpressing wild type TRPV6 and the G_660_R mutant (maternal genotype), as well as wild type TRPV6 and the R_510_stop mutant (paternal genotype) and G_660_R and R_510_stop mutant which reflects the affected child (Figure 1D and Appendix A). It can be seen that the combination of the expressed mutant TRPV6 variants strongly reduces the Ca^2+^ uptake of expressing cells. The peak value of the combination present in the affected child is 48% of the maternal and 51% of the paternal combination (Figure 1E). The experiment also shows that the reduced Ca^2+^ uptake is not an effect of the amount of functional TRPV6 channels, otherwise, one would expect to also see a reduced Ca^2+^ signal using the paternal combination (TRPV6 WT and R_510_stop mutant) which is not the case. To test that in the coexpressing experiments, both variants were synthesized and we expressed the combinations of constructs cloned in IRES-GFP vectors and in IRES-RFP vectors. Next, we asked if the amount of TRPV6 mutant proteins might be reduced in TRPV6 expressing cells as consequence of an unfolded protein response. Therefore, we expressed all constructs alone or as combinations that reflect the parents and the affected child. It can be seen on Western blots using two different TRPV6 specific antibodies that all constructs are present (Figure 1F). Thus, according to this experiment, there is no evidence that unfolded protein response/degradation occurs in the overexpressing cells. We also transfected the TRPV6 R_510_stop mutant alone and did not detect a higher Ca^2^ uptake, as seen in cells expressing the empty vector, which shows that the mutation completely abolishes the Ca^2+^ uptake of the mutated TRPV6 protein in expressing cells. It should be mentioned that the construct of the stop mutant contained the full length *TRPV6* cDNA in which the stop triplet was inserted. This experiment confirms the Western blot experiment and shows that in HEK293 cells the in frame R_510_stop codon present in the cDNA of *TRPV6* is not translated and leads to a truncated protein, as expected (~53 kDA). The experiment was performed to exclude that a read-through phenomenon occurs, as described by Li and Zhang [22].

#### 2.1.1. The G_660_R Mutation Can Be Rescued by Alanine

Next, we asked if the G_660_R mutant in combination with the R_510_stop mutant leads to a decreased Ca^2+^ uptake as the result of the positively charged amino acid arginine. If so, is it possible to obtain a rescuing effect by introducing an alanine residue instead (G_660_A mutation)? We coexpressed the G_660_A mutant with the R_510_stop mutant and measured Ca^2+^ signals comparable to the combination of wild type/R_510_stop mutant, as present in the father (Figure 2A). This result shows that the G_660_A mutation rescues the Ca^2+^ uptake. In addition, we replaced the G_660_ residue with another positive charged amino acid, resulting in a G_660_K mutation (lysine, Figure 2B). This mutation had a similar effect to the G_660_R mutation, if coexpressed with the truncated R_510_stop mutant. In addition, replacement by a negative charged amino acid, G_660_E, greatly reduced Ca^2+^ uptake (Figure 2C). We also tried to rescue the G_660_R mutation by introducing several mutations within the interacting sequence of the truncated R_510_stop mutation (Figure 2D described in detail below). The data indicate that, at position 660 of the human TRPV6 sequence, positive as well as negative charged amino acids affect the function of the channel when coexpressed with the truncated TRPV6 R_510_stop mutant. Therefore, one would expect that a noncharged amino acid at corresponding positions is strictly conserved within mammalian TRPV6 proteins and this is, indeed, the case. Although the G_660_A mutant rescued Ca^2+^ uptake, within all mammalian TRPV6 sequences the G_660_ residue is invariant (Figure 2E).

#### 2.1.2. Functional TRPV6 Channels Cannot Be Formed When the Subunits Are Expressed as Two Independent Parts

The TRPV6 stop mutant is characterized by the in frame stop codon which replaces R_510_; thus, the stop mutant corresponds to amino acid 1-509 of the TRPV6 protein. We made a TRPV6 construct in which amino acids 1-509 are not present but R_510_ was replaced by an artificial methionine, resulting in M_510_. This construct contains the amino acids M_510_-to I_765_, which represent the complete C-terminus, including the pore region of the TRPV6 protein. We coexpressed the latter construct with the R_510_stop mutant to test if cells can form functional TRPV6 channels (Figure 2F). We compared the coexpression with the single expression of TRPV6 M_510_-I_765_ but could not find significant differences. This indicates that functional TRPV6 channels cannot be formed from the R_510_stop mutant in combination with the TRPV6 M_510_-I_765_ construct. Next, we analysed the position of the particular G_660_ residue within the structure of the TRPV6 channel [18]. G_660_ is located at the boundary surface of the TRPV6 subunits in a large distance to the pore of the TRPV6 channel. The location implicates an influence of subunit assembly rather than parameters influencing the functionality of the pore directly. We suggest, from the TRPV6 structure, that the G_660_ that is located within the C-terminus of TRPV6 interacts with the N-terminal sequence of the adjacent TRPV6 subunit. 

#### 2.1.3. The G_660_R Mutation Cannot Be Rescued by Mutations in the N-Terminus of the Interacting Subunit

We identified amino acid residues within the N-terminus of TRPV6, to be considered as interaction partners of the G_660_ using the structural data published by Saotome and coworkers [18]. We identified, as a possible interacting sequence, a QQKR_83_ motif within the N-terminus of TRPV6, with K_82_ being at a distance of about 10.43Å to G_660_. We cloned a number of constructs, introducing one or two negatively charged amino acids in the C-terminus of the QQKR_83_ motif in the R_510_stop mutant, and coexpressed these constructs with the G_660_R mutant to see if a negative charged mutation in the truncated R_510_stop mutant can rescue the effect of the G_660_R mutation. We analysed four mutations within the QQKR_83_ sequence, namely, the mutations K_82_R, K_82_ER_83_E, Q_80_EQ_81_E and Q_80_E, which were cloned into the truncated R_510_stop mutant, and coexpressed these constructs with G_660_R mutant and tested if these can rescue the G_660_R mutation present in the full length protein. The expressed combinations of the four mutated truncated constructs showed a decreased Ca^2+^ signal and did not rescue the G_660_R mutation (Figure 2D, Appendix A).

#### 2.1.4. The TRPV6-R_510_stop Subunit Interacts with the Full Length TRPV6 Subunit

The previous experiment requires the assumption that the truncated stop mutant can still interact with the G_660_R mutant. This assumption is supported by the finding that the N-terminal ankyrin repeats which are important for the multimerization of the TRPV6 channel, are also present in the truncated mutant [23]. In addition, we performed a coimmunoprecipitation experiment, which shows that the truncated TRPV6 present in the child can interact with the full length TRPV6 protein (Figure 3A,B). We fused GFP to TRPV6 resulting in TRPV6-R_510_-GFP and TRPV6-GFP. As shown earlier by Hirnet and coworkers, the TRPV6 protein occurs as glycosylated and non-glycosylated protein [24]. The glycosylation site is located in between transmembrane S1 and S2, and is present in the full length TRPV6 as well as in the truncated TRPV6-R_510_-GFP variant. The TRPV6 protein was fished with the TRPV6 specific antibody 429, which binds to the C-terminus of TRPV6, and the co-immunoprecipitate (COIP) was analysed on a Western blot with a GFP antibody. Both proteins, TRPV6-R_510_-GFP and TRPV6-GFP, as well as glycosylated forms, were detected by COIP, which shows that both proteins interact. Another COIP experiment using TRPV6-G_660_R-RFP and TRPV6-R_510_-GFP fusion proteins also shows that the mutant variants present in the affected child can interact (Figure 3C,D). The interaction of both fusion protein was also confirmed by mass spectrometry (Figure 3E).

#### 2.1.5. The Amount of the Full Length TRPV6 Channel in the Plasma Membrane Is Reduced

Next, we asked if we can also detect the different TRPV6 subunits in the plasma membrane. We performed a biotinylation experiment, which shows that a small amount of the truncated TRPV6-R_510_stop variant is detectable in the plasma membrane of expressing cells (Appendix A). Furthermore, we expressed the combinations of TRPV6 WT and TRPV6-R_510_stop, as well as the combination of TRPV6-G_660_R and TRPV6-R_510_stop, and performed another biotinylation experiment. Although TRPV6 WT, as well as TRPV6-G_660_R, were present in the plasma membrane, it is clearly visible that the amount of the TRPV6-G_660_R in the plasma membrane is greatly reduced (Appendix A).

#### 2.1.6. The G_660_K Mutation Cannot Be Rescued by Mutating W_85_ of the Interacting Subunit

In the experiment shown in Figure 2D, we identified the N-terminal sequence of TRPV6, which is in close proximity to G_660_. Next, we emulated the effect of the G_660_R mutation in the affected child. The introduction of the R_660_ residue may lead to a slight shift in the interacting TRPV6 subunit and place the R_660_ residue next to a tryptophan residue, W_85,_ of the interacting subunit. To test if an exchange of W_85_ to alanine (W_85_A), arginine (W_85_R) or glutamate (W_85_E) can rescue the G_660_R mutation, we also coexpressed the latter three mutant constructs as truncated R_510_stop variants with the G_660_R mutation (Figure 4A,B). The three mutations, W_85_A, W_85_R and W_85_E, did not rescue the G_660_R mutation. The data indicate that the G_660_, which is strictly conserved among all mammalian TRPV6 proteins, is important for correct channel function when coexpressed with the TRPV6 R_510_stop mutant. In addition, we compared the paternal TRPV6 combination (TRPV6 and R_510_stop variant) with the combination of the child in the permanent presence of Ca^2+^ ions and measured the basic cytosolic Ca^2+^. It can be seen that the cytosolic Ca^2+^ level is significantly lower in cells mimicking the affected child (Figure 4C).

### 2.2. Comparative Proteome Analysis of Tissue Sections Obtained from a Healthy Placenta and the Placenta of the Affected Child

#### 2.2.1. HTRA1 and Cathepsin G Are Upregulated in the Placenta of the Affected Child

Since the malfunction of the mutated TRPV6 protein changes Ca^2+^ homeostasis in the placenta and leads to hyperparathyroidism, we analysed whether the dysfunction of the channel alters the protein expression profile in the placenta of the affected child. Therefore, we analysed sections of paraformaldehyde embedded placenta tissue from the sick and from a healthy child (with no skeletal dysplasia and hyperparathyroidism). After separation by gel electrophoresis, we analysed the extracted proteins using label free nano LC mass spectrometry. Three independent analyses were performed from each placenta. Using this approach, a total of 740 individual proteins were identified in both placentas (Figure 5A–C), 649 in the placenta of the child and 600 in the control placenta. As we showed previously, TRPV6 is mainly expressed in human syncytiotrophoblasts [1,25]. We used placental alkaline phosphatase (Swissprot: P05187 (PPB1_HUMAN) as a fetal syncytiotrophoblasts marker to evaluate the share of fetal syncytiotrophblast cells present in the tissue sections in both groups [26]. Combining the datasets for both categories (healthy vs. sick placenta), the mean amount of alkaline phosphatase was not different, which indicates that the content of syncytiotrophoblasts in both groups was similar. Next, we performed a label free semiquantitative analysis by counting the total number of peptide spectra belonging to the individual proteins of three datasets from both categories. Doing this, 15 proteins were exclusively or significantly more abundant in the affected placenta, while four proteins were downregulated compared to the healthy placenta (Figure 5A,B). Two proteases, high-temperature requirement A serine peptidase 1 (HTRA1) and cathepsin G, were only identified in the affected placenta. In a previous work, we analysed murine placenta trophoblasts from *Trpv6*^-/-^ and wild type mouse, and also found HTRA1 protease being upregulated in the TRPV6 deficient placenta [27].

#### 2.2.2. The HTRA4 Protease Is Upregulated in BeWo Cells in the Presence of Low Ca^2+^

Next, we tested if TRPV6 is expressed in a cell line, BeWo, which serves as model for throphoblast cells. After immunoprecipitation, we unambiguously identified several tryptic TRPV6 peptides, covering 15% of the human TRPV6 sequence (SwissProt: Q9H1D0) by mass spectrometry and Western blot (Figure 6A and Appendix A). With the confirmation that TRPV6 is expressed in these human cells, we cultured BeWo cells in the presence of high (0.65 mM) or low Ca^2+^ (0.35 mM), and analysed the protein expression again by mass spectrometry. By this approach, 2337 proteins were detected for both conditions: 2154 proteins were identified in lysates obtained after cultured in low Ca^2+^ and 1890 proteins were identified under high Ca^2+^ conditions. Interestingly, one serine protease, HTRA4, was detected to be five times more abundant in BeWo cells cultured using low Ca^2+^ (Figure 6A–D). Taken together, our Ca imaging experiments and the proteome analysis show that a disturbed assembly of the TRPV6 subunits in the placenta of the sick child leads to a massive reduction in the Ca^2+^ influx and, presumably, to a reduced Ca^2+^ content in trophoblasts, which, in turn, triggers a higher expression of serine proteases. The data may explain the phenotype of the observed child. The results raise the question of if TRPV6 is involved in the syncytialisation of the placenta. BeWo cells were treated with forskolin which induces syncytialisation and we analysed TRPV6 expression by Western blot (Appendix A). As an indicator for syncytialisation, we used a zonula occludens (ZO-1) specific antibody. TRPV6 was immunoprecipitated from the BeWo cells and detected using two different TRPV6 specific antibodies. The experiment indicates that TRPV6 expression is very similar in forskolin treated and untreated BeWo cells. On the other hand, we compared murine TRPV6 deficient trophoblasts with wild type trophoblasts and found that the percentage of cells that showed a distinct ZO-1 staining at the cell membrane is very low ( Appendix A), meaning that the trophoblast cells are fully syncytialised independent of the *Trpv6* genotype. The result implies that, at least in the murine trophoblasts, TRPV6 is not involved in the syncytialisation.

## 3. Discussion

The human TRPV6 protein is expressed in a few tissues, e.g., pancreatic acini and the trophoblast layer of the placenta. Several recent studies describe new born children who suffer from hyperparathyroidism with undermineralized bones [13,14,15,16,17]. However, the underlying cause of the disease seems to be the TRPV6 gene. Most of the identified TRPV6 mutations when cloned and overexpressed affect the function of the channel. Thus, typically, the Ca^2+^ influx via TRPV6 channels is greatly reduced. Interestingly, the parents of the described children were healthy at their time of birth, which indicates that one dysfunctional TRPV6 allele does not lead to a dramatic undermineralization of the skeleton. 

### 3.1. The TRPV6 Mutations of the Affected Child Lead to a Decreased Surface Expression of Functional Channels

We examined the role of two TRPV6 mutations in an affected child [13], in which one *TRPV6* allele contains a premature stop codon, R_510_stop, whereas the other allele contains, at position 660, a G to R (G_660_R) substitution. Surprisingly, the overexpressed G_660_R mutation behaves very similarly to the wild type TRPV6 channel, whereas, as expected, the truncated TRPV6 with the premature stop codon, R_510_stop, exhibits no detectable Ca^2+^ conductance. Coexpression studies that mimic the parental TRPV6 combinations, thus, wild type plus G_660_R or wild type combined with the truncated R_510_stop TRPV6, also did not reveal significantly reduced Ca^2+^ activity. Only the combination of the truncated R_510_stop TRPV6 with the G_660_R mutation showed a greatly reduced Ca^2+^ activity. The interpretation of the experimental data leads to the following model (Figure 7A–E): The maternal combination, TRPV6 WT allele and G_660_R mutation, does have a minor effect on the function of the channel. In addition, the paternal combination of the truncated R_510_stop mutation and a WT allele does not lead to reduced TRPV6 activity. Only the combination of the G_660_R and R_510_stop mutation shows a reduced Ca^2+^ uptake, most likely due to a disturbed subunit assembly and reduced surface expression. Interestingly, the particular glycine residue, G_660_, is conserved among all mammalian TRP-proteins, which emphasizes its indispensable importance for the correct function of the subunits. However, this becomes evident when the mutation was coexpressed together with the truncated TRPV6 R_510_stop mutant. The next related TRP protein, TRPV5, contains a glycine residue corresponding to the G660 of the human TRPV6 sequence, which is conserved in mammalian TRPV5 proteins. In addition, the interacting sequence within the N-terminus of TRPV6 is also conserved in TRPV5 proteins. We suggest that the G660 residue is critical for the interaction of N and C-termini of TRPV6 subunits. A mutation of this residue disturbs subunit assembly/membrane trafficking, and cannot be rescued by the tested mutations using the truncated TRPV6 R510stop constructs. The data provide the molecular basis of why these mutations lead to a pronounced undermineralization and, as a consequence, to the dysplasia of the skeleton of the affected child [13]. A recent publication shows that a small percentage of patients with nonalcohol dependent pancreatitis contain mutations that affect the TRPV6 gene [28]. All of the patients who were examined contained only one defective TRPV6 allele, which, when overexpressed, showed reduced Ca^2+^ activity [16]. In addition, mutations were found that lead to a dysfunctional channel, in which the closing/inactivation behaviour of the TRPV6 channel is affected. These combinations are thought to be potentially toxic as the result of Ca^2+^ overload. Suzuki and coworkers published a patient who contained the combination of a TRPV6 with reduced Ca^2+^ and one with an enhanced Ca^2+^ conductance [15]. This combination also leads to a reduced mineralisation of the skeleton in this patient. In addition, the overexpression of TRPV6 transcripts seems to be critical. Thus, overexpression of *TRPV6* transcripts is associated with several malignancies, which include cancer derived from prostate and breast. The present work shows that the G_660_R mutation, which does not alter the function of the channel when expressed as homomultimer, is critical for the mineralization of the bones when coexpressed in combination with the R_510_stop mutation. We suggest that these mutations affect the assembly of the subunits and lead to the reduced surface expression of functional TRPV6 channels. It can be seen in Appendix A that, in the plasma membrane fraction of the paternal combination, the amount of TRPV6 WT is higher compared to the combination which reflects the affected child. Thus, the amount of TRPV6-G_660_R seems to be greatly decreased. On the other hand, in both cotransfections (paternal and child), occurs an additional protein that is detectable with the N-terminal ab 1271 but not with the C-terminal ab 429. We speculate that this protein is a C-terminal breakdown product of the TRPV6 channel. Most notably, the amount of the breakdown product is higher in the plasma membrane fraction of the affected child. The data are conclusive that the occurrence of a lower amount of TRPV6 channels results in the decreased calcium uptake of expressing cells, and may explain the undermineralization of the skeleton of the affected child.

### 3.2. Loss of TRPV6 Function Is Associated with the Upregulation of Placental Proteases

In addition, we show that the loss of TRPV6 is associated with changes in the expression level of a few proteases. We compared the protein profiles of a healthy placenta with the placenta of the affected child and found that two proteases, HTRA1 and cathepsin G, were only detectable in the placenta of the affected child. A link between cathepsin G and Ca^2+^ transport has been reported by Peterson and coworkers [29]. The authors show that, in cultures of endothelial/epithelial cells, cathepsin G affects the formation of intercellular gaps and thereby increases the permeability for Ca^2+^ ions. The underlying mechanism is not known but the expression of cathepsin G in the placenta of the sick child may act as a compensation mechanism to increase the Ca^2+^ uptake/transfer to the foetus, which is reduced as a consequence of the mutations in the TRPV6 channel. 

Additionally, the serine protease HTRA1was exclusively detected in the placenta of the affected child. HTRA1 was also more abundant in murine *Trpv6*-deficient placenta trophoblasts [27]. In line with this observation, murine *Trpv6* deficient placenta contains a higher activity of HTRA1, which leads to enhanced degradation of extracellular matrix proteins, such as fibronectin. This also applies to the human trophoblast derived cell line BeWo, in which TRPV6 is endogenously expressed. In the presence of low Ca^2+^ in the medium, which mimics the loss of TRPV6 channels, BeWo cells overexpress the protease HTRA4, which is closely related to the HTRA1 protease. The HTRA1 protease is preferentially expressed in human placenta during the third trimester, is localized in syncytiotrophoblasts and cytotrophoblast intracellularly and is also detectable extracellularly [30]. Both proteases, HTRA1 and HTRA4, are elevated in the placenta of preeclampsia patients [31]. In this study, the overexpression of HTRA1 and HTRA4 in the human trophoblast cell line HTR-8 reduces cell migration and, under hypoxic cell culture conditions, HTRA1 and HTRA4 expression increases. In line with results reported for HTR-8 cells, we detected a 5-fold increase in HTRA4 expression in the presence of low Ca^2+^ concentration in the medium, which obviously induces a comparable stress in BeWo cells. 

In addition, in pancreatic acini, which also express TRPV6, there is a link between protease activity and dysfunctional TRPV6 channels. A subpopulation of patients with chronic pancreatitis shows mutations in one *TRPV6* allele [28]. The same study also shows that *Trpv6* knock in mice, which express non Ca^2+^ permeable channels, are more sensitive to treatment with cerulein. After cerulein treatment, the serum level of a few pancreatic enzymes, namely, amylase and lipase, are accelerated in this mouse model. A link between the onset of pancreatitis and the intracellular Ca^2+^ concentration of rodent acinus cells was observed earlier [32]. Taken together, we demonstrate that the mutations in the *TRPV6* gene of the affected child lead to reduced Ca^2+^ uptake in heterologous expression system, which may explain the skeletal dysplasia observed in the new born. In addition, the Ca^2+^ deficiency seems to be connected to enhanced expression of proteases in the placenta of the affected child.

## 4. Materials and Methods

### 4.1. Cloning

Mutations were introduced using Fusion polymerase (NEB, Ipswich, MA, USA) and a plasmid which contains the *TRPV6* cDNA cloned in pCDNA3. Mutations in the C-terminal part replacing G_660_ were subcloned as BstEII and MfeI fragments in TRPV6-pCAGGS-IRES-eGFP and TRPV6-pCAGGS-IRES-mRFP. To generate an in frame stop codon replacing R_510_, we used a similar strategy. We subcloned mutations within the QQKR_8 3_as well as the W_85_ present in the N-terminus of TRPV6 as PshAI-BestEII fragments in the appropriate vectors.

### 4.2. Immunoprecipitation of TRPV6 from Transfected HEK293 and BeWo Cells

BeWo cells were grown to 90% confluence in cell culture dishes (282cm^2^ each) and harvested using 10 mL PBS and a cell scraper, cells were sedimented (1400× *g*, 5 min) and washed with PBS. The cell pellet was resuspended in 1 mL RIPA buffer (150 mM NaCl, 50 mM Tris HCl, pH 8.0, 5 mM EDTA, 1% Nonidet P40, 0.1% SDS, 0.5% Na-deoxycholate, pH 7.4), supplemented with proteinase inhibitors (Roche, Mannheim, Germany). Cell solution was sheared ten times (27G gauge needle) on ice and then incubated for 30 min at 4 °C on a shaker. After centrifugation at 100,000× *g* at 4 °C for 45 min, the supernatant containing the solubilized proteins was incubated for 16 h at 4 °C in the presence of 10 µg anti-TRPV6 antibody 429 (directed against the c-terminus) coupled to 50µL of Dynabeads™ Protein G (Invitrogen, Schwerte, Germany). The beads were collected using a magnetic rack, washed three times with 1 mL RIPA buffer and were eluted with 50 µL denaturing sample buffer (final concentration: 60 mM Tris HCl, pH 6.8, 4% SDS, 10% glycerol including 0.72 M beta-mercaptoethanol). The elution was incubated for 20 min at 60 °C and analysed by mass spectrometry. To co-immunoprecipitate TRPV6 from HEK293 cells two dishes were cotransfected with WT TRPV6 and truncated TRPV6 (R_510_stop mutant) GFP fusion constructs. 

The stop mutant was fused to GFP to discriminate this protein from IgGs on the Western blot. Harvested cells were solved in 1.5 mL lysis buffer (TBS including 1% digitonin and protease inhibitors). After 10× shearing, the cells were incubated for 1 h at 4 °C on a shaker device and centrifuged at 100,000× *g* for 45 min at 4 °C. The cell lysate was incubated for 2 h at 4 °C with antibody 429 coupled to magnetic protein A/G beads (ThermoFisher, Karlsruhe, Germany). Antibody 429 is directed against the C-terminus of TRPV6 and binds only WT TRPV6 but not the R_510_stop mutant. The beads were washed 3 times using 1ml of 0.1% digitonin buffer including protease inhibitors, denatured in 60 µL sample buffer, subjected to SDS-PAGE electrophoresis following Western blot procedure as described below. The Western blot was incubated with the GFP antibody.

### 4.3. Surface Biotinylation and Western Blot 

The method has already been described (Fecher-Trost et al., 2013). A 75cm^2^ flask with confluent COS cells was transfected with the appropriate constructs and cultured for 24 h, washed twice with ice-cold phosphate buffered saline (137 mM NaCl, 2.7 mM KCl, 10 mM Na_2_HPO_4_, 2 mM KH_2_PO_4_) containing 1 mM MgCl_2_ and 0.5 mM CaCl_2_ (PBSB, pH, 8.0), and incubated in the presence of NHS-LC-biotin freshly diluted in PBSB at 0.5 mg/mL for 30 min at 4 °C. The reaction was stopped by washing twice with PBSB containing 0.1% (*w*/*v*) bovine serum albumin and once with PBS, pH 7.4. Cells were harvested from the flasks by shaking in PBS supplemented with 2 mM EDTA. The cells were centrifuged at 1000× *g* at 4 °C for 5 min and resuspended in ice cold lysis buffer (PBS containing 1% Triton X-100, 1 mm EDTA, and a mixture of protease inhibitors). Cell lysates were rotated at 4 °C for 30 min to solubilize proteins; after centrifugation at 1000× *g* and 4 °C for 5 min, the amount of protein was determined using BCA (ThermoFisher, Waltham, MA, USA), and the protein solution (~1 mg) was added to 100 μL of avidin–agarose beads pre-equilibrated in lysis buffer and incubated at 4 °C for 2 h. The biotin–avidin–agarose complexes were washed 4 times with lysis buffer supplemented with 0.25 mM NaCl. Biotinylated proteins were eluted in 100 μL of 2-times denaturing electrophoresis sample buffer and incubated at 60 °C for 30 min before SDS-polyacrylamide gel electrophoresis (SDS-PAGE) on 8% Bolt gels, (Invitrogen, Carlsbad, CA, USA) in a Bolt buffer system. Cell lysate input corresponds to protein samples taken before adding avidin–agarose beads. The proteins were electrophoretically separated, blotted, and probed with TRPV6 N-terminal ab 1271 and C-terminal ab 429, respectively. The endoplasmic reticulum protein calnexin was used as a control.

### 4.4. Antibodies

The following in house generated anti-TRPV6 antibodies were used: Polyclonal antibody 1271 and 429 directed against N- and C-terminus of human TRPV6 [24,25], respectively and monoclonal YFP/GFP antibody [25]. All antibodies were affinity purified before use. Commercial ZO1 antibody was from Invitrogen. 

### 4.5. Calcium Imaging 

Intracellular live cell Ca^2+^-imaging experiments were performed using a Polychrome V and CCD camera (TILL Imago)-based imaging system from TILL Photonics (Martinsried, Germany) with a Zeiss Axiovert S100 fluorescence microscope equipped with a Zeiss Fluar 20×/0.75 objective. Data acquisition was accomplished with the imaging Live Acquisition software (TILL Photonics). Data were analysed using the Offline analysis software (TILL Photonics). Cells were incubated in media supplemented with 4 μm Ca^2+^-sensitive fluorescent dye Fura-2-AM (Molecular probes, Eugene, USA) for 30 min in the dark at room temperature and washed 4 times with nominally Ca^2+^ free external solution (140 mM NaCl, 5 mM KCl, 1 mM MgCl_2_, 10 mM HEPES, 10 mM glucose, adjusted to pH 7.2 with NaOH) to remove excess Fura-2-AM. The Fura-2-AM loaded cells, growing on 2.5 cm glass coverslips, were transferred to a bath chamber containing nominally Ca^2+^ free solution, and Fura-2 fluorescence emission was monitored at >510 nm after excitation at 340 and 380 nm for 30 ms each at a rate of 1 Hz for 600 s. Cells were marked, and the ratio of the background-corrected Fura-2 fluorescence at 340 and 380 nm (F340/F380) were plotted versus time. After reaching a stable F340/F380 base line, 2.5 mM CaCl_2_ was added to the bath solution, and cytosolic Ca^2+^-signals were measured. 

### 4.6. Cell Culture and Transfection of HEK293 Cells

HEK293 cells were grown in culture dishes (3 cm diameter) with poly l-lysine-coated glass coverslips (diameter 2.5 cm) until 80% confluence and then transiently transfected with 2.5 μg of appropriate cDNA constructs in 5 mL of Lipofectamin 3000 (ThermoFisher, Karlsruhe, Germany). For Fura-2-AM measurements, cells were transfected with TRPV6 constructs cloned in pcAGGS-IRES-GFP or IRES-mRFP vectors. Cotransfection was carried out with a combination of the appropriate constructs cloned in vectors with green and red fluorescent proteins (1.25 µg each). Coverslips with transfected cells were used for Ca^2+^ imaging experiments 24 h after transfection. 

### 4.7. Modelling G_660_R Mutation

We used the RCSB PDB [33] software and the rat TRPV6 structure (PDB ID: 5IWK) to identify amino acids in close proximity to the glycine residue, which corresponds to G_660_ in the human TRPV6 sequence [18]. It was concluded that the C-terminal part of wild type TRPV6 interacts with the amino acid sequence QQKR_83_ present within the N-terminus of the interacting TRPV6 subunit. In addition, we analysed the TRPV6 structure in which the R_660_ mutation was modelled. The R_660_ residue comes in close proximity to the tryptophan residue W_85_ present in the N-terminus of the interacting subunit.

### 4.8. Protein Extraction from Paraformaldehyde/Formalin Fixed Placenta Tissue Sections

For protein extraction, two to three unstained slides of 4% formalin fixed placenta tissue (3µm) were scraped from glass microscope slices with a scalpel and resuspended in 80 µL protein extraction buffer (60 mM Tris, pH 6.8, 1.2 M glycerol, 0.78 M β-mercaptoethanol, 70 mM SDS, 10 mM arginine). Samples were consecutively incubated for 15 min on ice, 20 min at 100 °C and for 1 h at 80 °C. Protein extracts were centrifuged for 15 min at 14,000× *g* (4 °C) and the supernatant was transferred to a fresh vial. 60 µL per sample was loaded on a 4–12% BoltTM gel (ThermoFisher, Waltham, MA, USA) and electrophoresed. The gel was fixed and stained with a colloidal Coomassie, 16 gel bands/lane were isolated and digested using trypsin as described before [25].

### 4.9. BeWo Cell Culture

Human BeWo cells (ATCC^®^ CCL¬98™) were cultured in 3.5 cm diameter cell culture dishes (Corning, Tewksbury, USA), with culture medium (F-12 Nut Mix with 2 mM glutamax (ThermoFisher, Karlsruhe, Germany), 10% FKS (Corning, Tewksbury, MA, USA) and 1% penicillin/streptomycin (Sigma-Aldrich-Merck, Darmstadt, Germany). The cells grew in the presence of 5% CO_2_ at 37 °C. Cells were trypsinized, seeded in fresh cell culture dishes and cells grew until 40% of confluency was reached. The medium was changed after 24 h to medium containing 0.65 mM Ca^2+^ or to medium containing 0.35 mM Ca^2+^ including EDTA. Calcium concentration was determined using a Dri-Chem NX500i System (FujiFilm, Japan). The amount of EDTA was calculated using the WEBMAXCSTANDARD7/3/2009 software (https://somapp.ucdmc.ucdavis.edu/pharmacology/bers/maxchelator/webmaxc/webmaxcS.htm retrieved on 31 March 2021). Cells were cultured for additional 48h in either 0.65 mM Ca^2+^ or 0.35 mM Ca^2+^ medium, washed 3 times with PBS and removed from the dish with a cell scraper (Corning, Tewksbury, MA, USA) and resuspended in denaturing sample buffer. To prepare a sample for one mass spectrometry experiment, two cell culture dishes of the same medium condition were pooled. To induce syncytialisation of BeWo cells 200,000 cells were cultured in a flask (Falcon, 75 cm^2^) and medium was supplemented with 0.3% DMSO or 20 µM forskolin/0,3% DMSO for 48 h. Cells were stained with azur/eosin or with a ZO1-antibody. 

### 4.10. Preparation of BeWo Cell Lysates for Proteome Analysis

20 µL of BeWo cell lysates grown in 0.65 mM Ca^2+^ or 0.35 mM Ca^2+^ medium were separated on a NuPAGE^®^ 4–12% gradient gels (ThermoFisher, Karlsruhe, Germany). Proteins were fixed in the presence of 40% ethanol and 10% acetic acid and visualized with colloidal Coomassie stain (20% (*v*/*v*) methanol, 10% (*v*/*v*) phosphoric acid, 10% (*w*/*v*) ammonium sulfate, and 0.12% (*w*/*v*) Coomassie G-250). Fourteen gel pieces were cut/sampled and trypsin digested as described [25].

### 4.11. Primary Mouse Trophoblast Cell Culture

Trophoblasts from *WT* and *Trpv6^-/-^* placentae were isolated at E13.5 as previously described by Winter et al., 2020 [27]. 200,000 cells were seeded in a 3.5 cm dish on four uncoated glass coverslips (Orsa^tec^, 1.2 cm) and incubated with 2 mL medium (DEMEM, Gibco®, 10% FCS, 100 U/mL penicillin, 100 µg/mL streptomycin) at 37 °C and 5% CO_2_. After 24 h the medium was changed and cells were cultured for additional 4 days followed by ZO1-antibody staining. 

### 4.12. Zona Occludens 1 (ZO-1) Antibody Staining of Trophoblast and BeWo Cells

Coverslips with BeWo or mouse trophoblast cells were removed and washed with PBS. Fixation and permeabilization was performed by incubating the cells with methanol (−20 °C) on ice for 20 min. Cells were washed 4 times with PBS and blocking buffer (3% BSA, 1% normal goat serum and 0.1% Triton X100 in PBS) was added for 1 h at RT. After washing 4 times with PBS, cells were incubated over night at 4 °C in a buffer (1% BSA in PBS) containing a ZO-1 specific primary antibody (1:1000, Invitrogen). Cells were washed with PBS 4 times and incubated with an Alexa-fluor 488 anti-rabbit antibody (Invitrogen) and DAPI (2 µg/mL) in PBS. Cells were washed with PBS 4 times and mounted using Immu-Mount^TM^ (ThermoSCIENTIFIC). Stained BeWo cells were analyzed by using an Imager.M2 microscope (Zeiss) obtaining a Axiocam MRm (Zeiss). Trophoblast cells were analyzed using a Slightscanner (Axio Scan.Z1, Zeiss). The whole set of pictures was processed with the software Imaris (Oxford Instruments) and the number of nuclei were detected automatically. ZO-1 positive cells were counted manually and the percentage of ZO-1 positive cells was calculated.

### 4.13. Mass Spectrometric Measurement (Nano-LC–MS/MS)

Six µL of tryptic digested peptides derived from each gel piece (BeWo cell lysate, IPs from BeWo cells or human placenta lysates) were analysed by nano LC–ESI–MS/MS analysis using the set up (Ultimate 3000 RSLC nano system equipped with an Ultimate3000 RS autosampler coupled to an LTQ Orbitrap Velos Pro, (ThermoFisher, Dreieich, Germany)). Peptides were trapped on a C18 trap column (75 µm × 2 cm, Acclaim PepMap100C18, 3 µm, Dionex) and separated on a reversed phase column (nano viper Acclaim PepMap capillary column, C18; 2 µm; 75 µm × 50 cm, Dionex) at a flow rate of 200 nL/min with buffers A (water and 0.1% formic acid) and B (90% acetonitrile and 0.1% formic acid) using a 94 min gradient (BeWo cell lysates) or 120 min gradient (human placenta lysates). The effluent of the chromatography was sprayed into the mass spectrometer through a coated emitter (PicoTipEmitter, 30 µm, New Objective, Woburn, MA, USA) and ionized at 2.2 kV. MS spectra were acquired in a data dependent mode. For the collision induced dissociation (CID) MS/MS top10 method, full scan MS spectra (m/z 300–1700) were acquired in the Orbitrap analyser using a target value of 10^6^. Peptide ions with charge states >2 were fragmented in the high-pressure linear ion trap by low-energy CID with normalized collision energy of 35%.

### 4.14. Raw Mass Spectrometrical Data Analysis

The fragmented tryptic peptides were identified using the MASCOT algorithm and TF Proteome Discoverer 1.4 software (ThermoFisher, Waltham, MA, USA). Peptides were matched to tandem mass spectra by Mascot version 2.4.0 by searching of a SwissProt database (version2018_03, number of protein sequences, taxonomy human: 20,387). Data were analysed as described by Winter and co-workers [27].

## Figures and Tables

**Figure 1 ijms-22-12694-f001:**
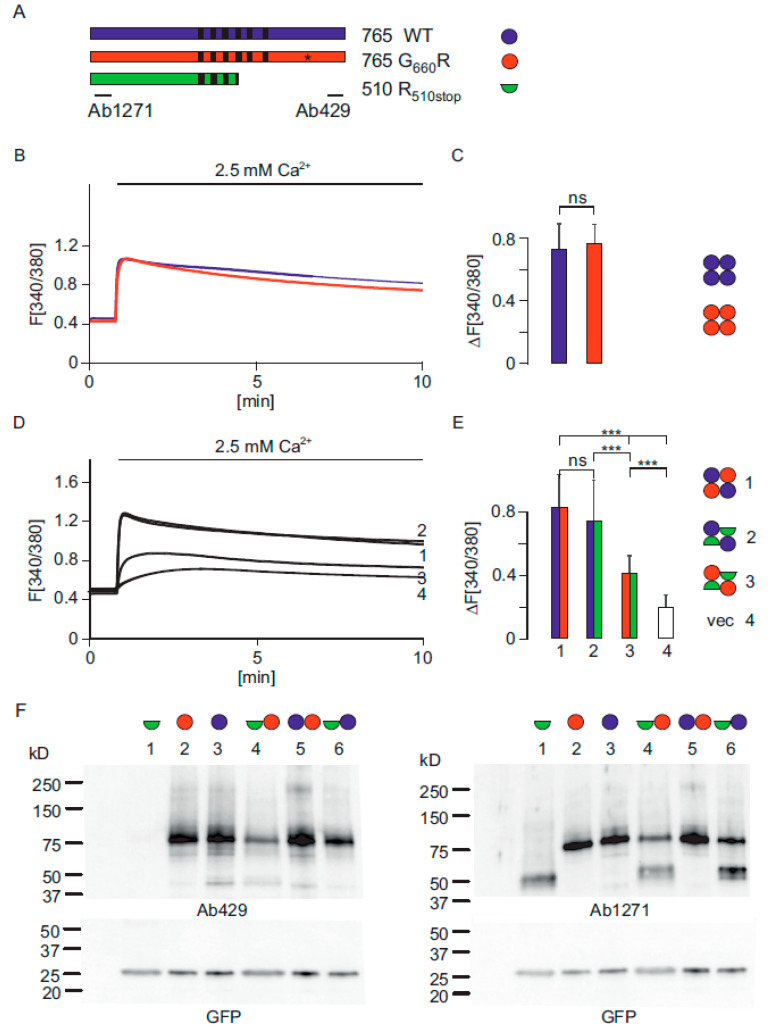
TRPV6 activity is reduced in HEK293 cells expressing mutant TRPV6 subunits present in the affected child. (**A**) TRPV6 constructs used for Ca^2+^ imaging and Western blots, TRPV6 WT (blue), TRPV6-G_660_R (red) and TRPV6 R_510_stop (green). Transmembrane domains (black bars), G_660_R mutation (*) and binding sites for TRPV6 specific antibodies 1271 and 429 are indicated. (**B**,**C**) Ca^2+^ imaging of TRPV6 WT (blue, n/N = 132/3) and TRPV6-G_660_R (red, n/N = 126/3) in HEK293 cells and statistical analysis of the peak values. Circles indicate TRPV6 subunits. (**D**,**E**) Coexpression of TRPV6 WT-I-GFP and TRPV6-G_660_R-I-RFP, which reflects the maternal TRPV6-genotype (1, blue/red, n/N = 117/3), coexpression of TRPV6 WT-I-RFP and TRPV6-R_510_stop mutant I-GFP, which reflects paternal genotype (2, blue/green, n/N = 72/3), coexpression of TRPV6-G_660_R-I-RFP and TRPV6-R_510_stop mutant I-GFP, which reflects the child (3, red/green, n/N = 87/3), vector control (4, white, n/N = 82/2) and statistical analysis of peak values. n/N = cells/experiments. Asterisks assign significance differences (*** *p* < 0.001, ns = not significant). (**F**) Western blots of cells expressing TRPV6 constructs in HEK293 cells: lane1 TRPV6-R_510_stop mutant (green semicircle), lane2 TRPV6-G_660_R mutant (red circle), lane 3 TRPV6 WT (blue circle), lane 4 coexpression of TRPV6-R_510_stop and G_660_R mutants, lane 5 coexpression of TRPV6 WT and G_660_R mutant, lane 6 coexpression TRPV6 WT and TRPV6-R_510_stop mutant. All TRPV6 variants were expressed as I-GFP constructs. Western blot was probed with antibody 429 (left) and antibody 1271 (right). GFP control below.

**Figure 2 ijms-22-12694-f002:**
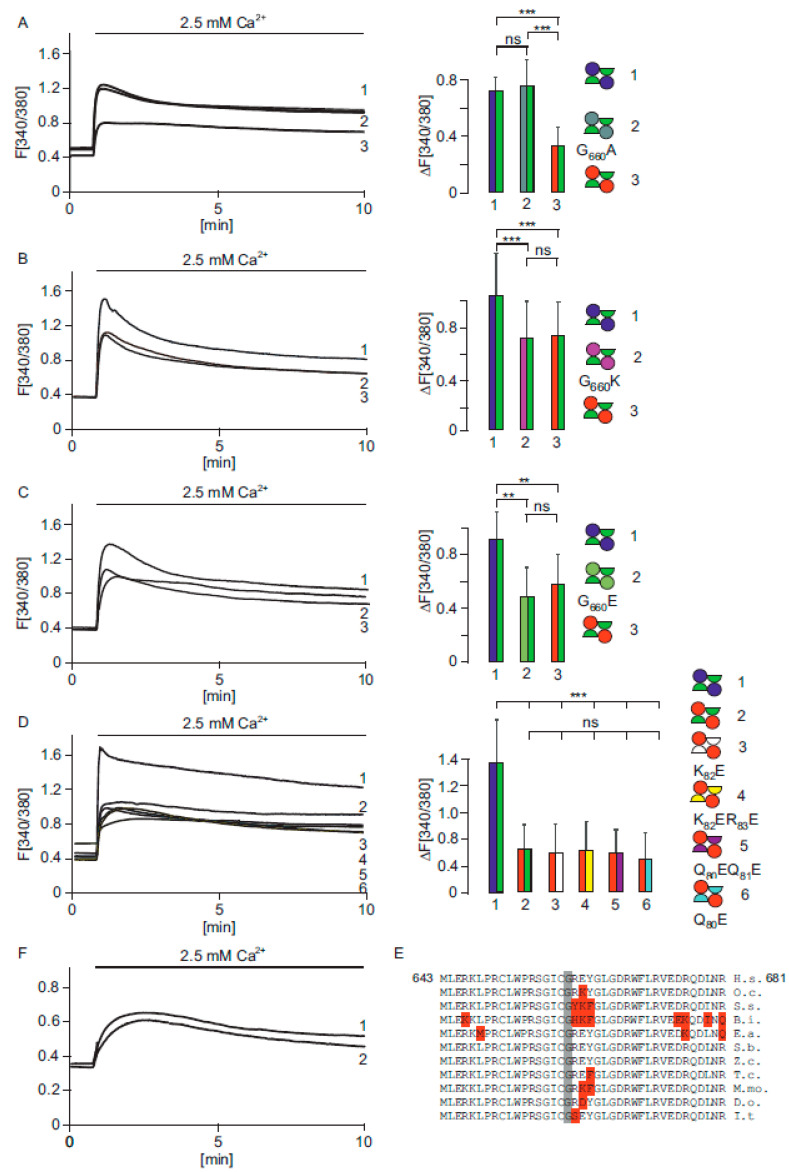
A TRPV6-G_660_A mutation rescues the G_660_R mutation. (**A**) Calcium imaging of cells coexpressing TRPV6 WT-I-GFP and TRPV6-R_510_stop-I-RFP (1, blue/green, paternal, n/N = 95/3), TRPV6-R_510_stop mutant I-RFP and TRPV6-G_660_A-I-GFP (2, red/grey, n/N = 97/3) and TRPV6-G_660_R I-GFP and TRPV6-R_510_stop mutant I-RFP (3, green/red, child, n/N = 91/3) n/N = cells/experiment. (**B**) Coexpression of TRPV6-R_510_stop mutant I-RFP and TRPV6-G_660_K-I-GFP (2, pink/green, n/N = 106/3) compared with the parental combination (1, blue/green, n/N = 79/3) and the child (3, red/green, n/N = 84/3). (**C**) Similar experiment as shown in (**B**), coexpression of TRPV6-R_510_stop mutant I-RFP and TRPV6 G_660_E-I-GFP (2, light green/green n/N = 55/3), compared with the parental combination (1, blue/green n/N = 65/3) and the child (3, red/green n/N = 44/3). (**D**) Coexpression of the TRPV6-G_660_R-I-RFP with several TRPV6-R_510_stop mutants cloned in I-GFP vectors, which, in addition, contain a second mutation within the N-terminal located sequence QQKR_83_. This sequence interacts with the C-terminal sequence in which the G_660_ residue is located. The following mutants were tested: K_82_E (3, red/white, n/N = 50/3), K_82_ER_83_E (4, red/yellow, n/N = 44/3), Q_80_EQ_81_E (5, red/magenta, n/N = 65/3), and Q_80_E (6, red/light blue, n/N = 45/3). The mutants were compared with the parental combination (1, blue/green, n/N = 52/3) and the child (2, red/green, n/N = 47/2). Here, n/N = cells/experiments. Asterisks assign significance differences (** *p* < 0.01, *** *p* < 0.001, ns = not significant). (**E**) Alignment of mammalian TRPV6 protein sequences from amino acid 643 to 681. G_660_ is strictly conserved (grey). H.s., Homo sapiens; O.c., Oryctolagus cuniculus; S.s., Sus scrofa; B.i., Bos indicus; E.a., Equus asinus; S.b., Saimiri boliviensis; Z.c., Zalophus californianus; T.c., Tupaia chinensis; M.mo., Monodon Monoceros; D.o., Dipodomys ordii; I.t., Ictidomys tridecemlineatus. (**F**) Expression of artificial TRPV6 construct which contains amino acids 510 to 765 (1, n/N = 117/3) and coexpression with the same construct and the TRPV6-R_510_stop mutant (2, n/N = 70/3).

**Figure 3 ijms-22-12694-f003:**
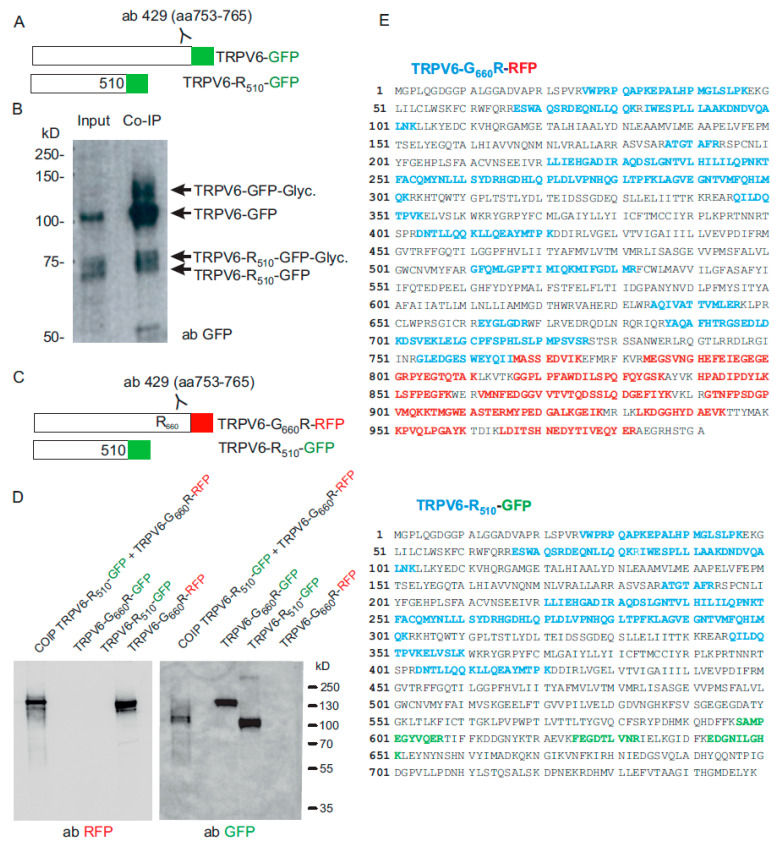
(**A**) TRPV6 fused to GFP (TRPV6-GFP) and TRPV6-R510 fused to GFP (TRPV6-R510-GFP, stop codon removed) were cotransfected in HEK293 cells and immunoprecipitated with a C-terminal TRPV6 specific antibody 429 (directed against aa 753-765 of TRPV6). (**B**) Western blot of the input and eluate from co-immunoprecipitaion (COIP) with a GFP antibody. (**C**) TRPV6-G660R fused to mRFP (TRPV6-G660R-RFP) and TRPV6-R510 fused to GFP (TRPV6-R510-GFP) were cotransfected in HEK293 cells. TRPV6-G660R-RFP was immunoprecipitated with TRPV6 antibody 429. (**D**) Detection of fused RFP and GFP tagged TRPV6-G660R and TRPV6-R510 proteins in cell lysates from single transfections and in the eluate obtained after cotransfection/co-immunoprecipitation (COIP). (**E**) Mass spectrometrical identification of TRPV6-G660R-RFP and TRPV6-R510-GFP proteins in the eluate of the COIP (as presented in (**C**,**D**)). Location of tryptic peptides identified by MS/MS fragmentation; TRPV6 (blue), RFP (red) and GFP (green).

**Figure 4 ijms-22-12694-f004:**
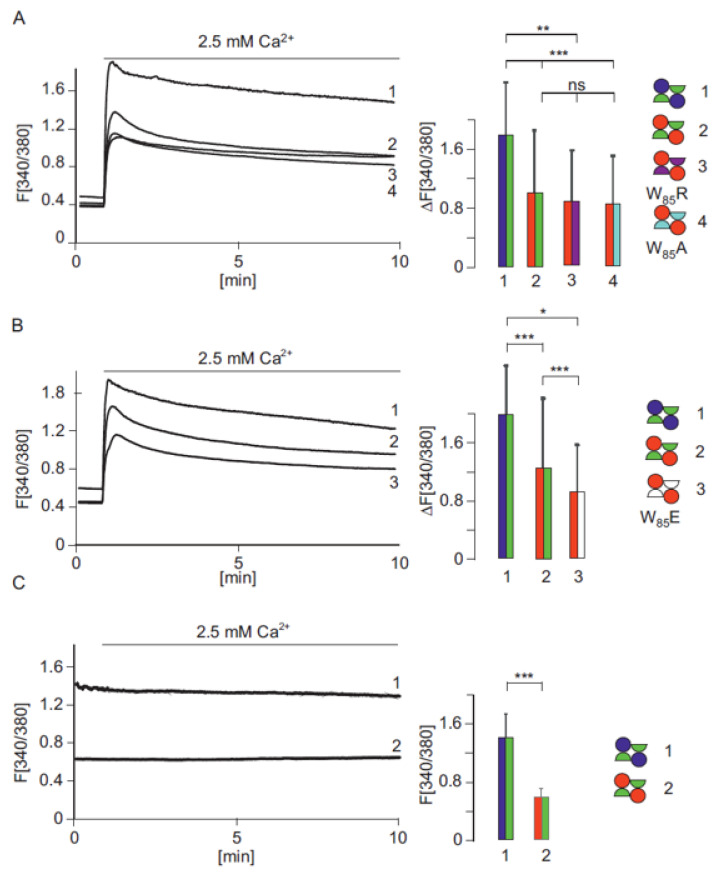
(**A**,**B**) Coexpression of TRPV6-G660R-I-RFP with several TRPV6-R510stop mutants which, in addition, contain W85 mutations. The tryptophan W85 was mutated to W85R (3, red/magenta, n/N = 47/3), W85A (4, red/light blue, n/N = 92/3), W85E (3 shown in B, red/white, n/N = 98/3) and compared with the TRPV6 combination present in the father (1, blue/green, n/n/N/N = 61/3, 55/3) and the child (2, red/green, n/n/N/N = 60/3, 120/3), respectively. (**C**) Expression of the TRPV6 combination of parental (1, blue/green, n/N = 48/3) and child (Red/green, n/N = 73/3) after loading FURA-2AM and incubation of the cells in the permanent presence of 2.5 mM Ca2+. n/N = cells/experiments. Asterisks assign significance differences (* *p* < 0.05, ** *p* <0.01, *** *p* < 0.001, ns = not significant).

**Figure 5 ijms-22-12694-f005:**
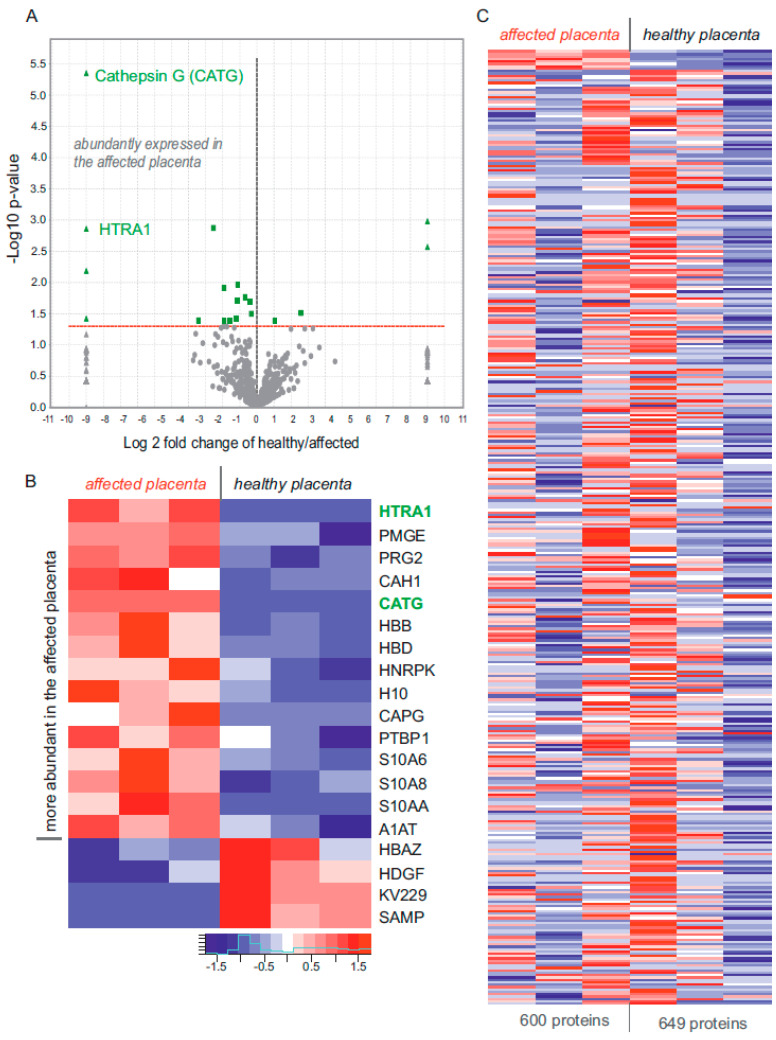
Proteome analysis of paraformaldehyde fixed tissue of a healthy placenta and the placenta of the affected child. (**A**) Vulcano-blot of a semiquantitative analysis of differentially expressed proteins identified by mass spectrometry of pooled tissue sections of a healthy placenta and the placenta of the affected child (*n* = 3 samples/genotype). Fifteen proteins are upregulated and four proteins downregulated in the placenta of the affected child. Up- and downregulated proteins were identified based on at least 1.3-fold changes in the total spectrum counts, with *p*-values < 0.05 using unpaired two-tailed Student’s t-test. Cathepsin G (CATG) and serine protease HTRA1 (green) are only detectable in the affected child. (**B**) Heatmap of Z-scores calculated from the total peptide spectra counts of proteins (UniProt Identifier), which were more abundant in the child (green triangles and squares shown in (**A**)). (**C**) Heatmap of identified proteins on the basis of total spectrum count values (shown as Z-scores) from three independent mass spectrometry samples prepared from placentas from healthy control and affected child (*n* = 3). In total, 740 proteins were identified by a least two unique peptides/protein of the healthy placenta and the placenta of the affected child.

**Figure 6 ijms-22-12694-f006:**
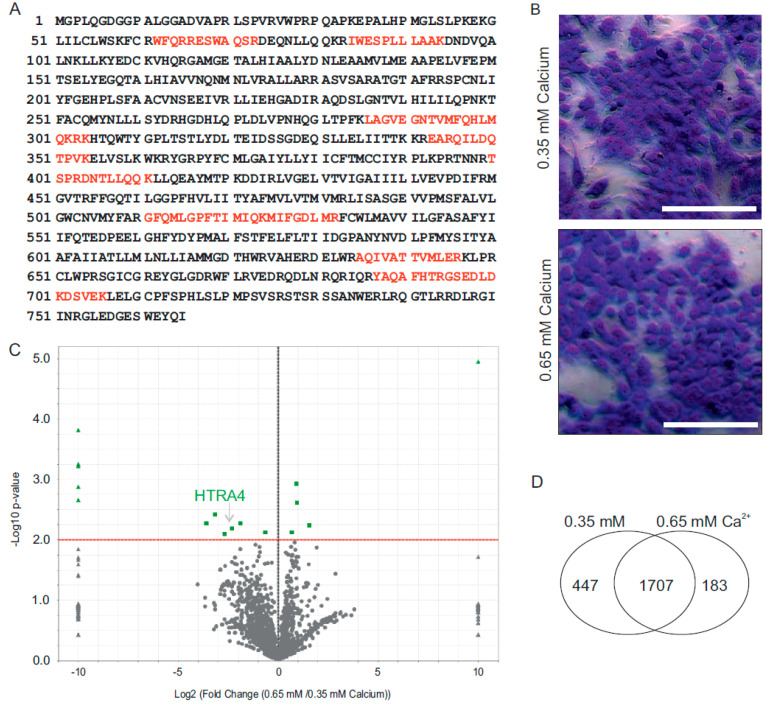
Proteome analysis of BeWo cells cultured in the presence of different Ca^2+^ concentrations. (**A**) Tryptic TRPV6 peptides identified by mass spectrometry after immunoprecipitation from BeWo cells (red) with TRPV6 ab 429. (**B**) BeWo cells were cultured in the presence of 0.65 mM (high) or 0.35 mM Ca^2+^ (low). Cells were fixed and stained with eosin/azur, scale bar: 200 µm. (**C**) Total protein identifications in BeWo cells cultured in the presence of high or low Ca^2+^. (**D**) Vulcano blot shows semi quantitative analysis of differentially detected proteins identified in BeWo cell lysates. Proteome analysis of the abundance of peptide spectra detected in BeWo cells in the presence of high or low Ca^2+^. Up- and downregulated proteins were identified based on at least 2-fold changes in the total peptide spectra abundance detected in three independent experiments with a *p*-value < 0.01, calculated using the unpaired two-tailed Student’s *t* test. Serine protease HTRA4 is more abundant in BeWo lysates cultured in 0.35 mM Ca^2+^.

**Figure 7 ijms-22-12694-f007:**
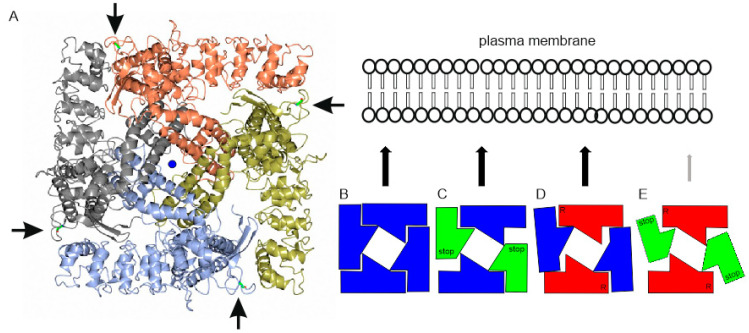
Putative assembly of TRPV6 subunits. (**A**) Crystal structure of rat TRPV6 (PDB ID: 5IWK, Saotome et al., 2016). View from the top. G_660_ is located at the border of the TRPV6 subunits (arrows). The pore of the channel is shown in the middle, a Ca^2+^ ion is indicated (blue). (**B**) Assembly of TRPV6 wild type subunits (blue). (**C**) Assembly of wild type and R_510_stop TRPV6 subunits, as present in the father (blue/green). (**D**) Assembly of wild type and the G_660_R TRPV6 subunit, as present in the mother (blue/red). (**E**) Assembly of truncated R_510_stop and G_660_R TRPV6 subunits, as present in the affected child (red/green). Ca^2+^ uptake is only decreased in the child because the abundance of functional TRPV6 channels in the plasma membrane is greatly reduced (indicated by grey arrow).

## Data Availability

Data is contained within the article or Appendix A.

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
