# Peer review of "Mutations That Affect the Surface Expression of TRPV6 Are Associated with the Upregulation of Serine Proteases in the Placenta of an Infant"

_ijms, 2021, doi:10.3390/ijms222312694_

Round 1
Reviewer 1 Report
Review report
Fecher-Trost et al. here investigated the role of two mutations in TRPV6 channels in the development of skeletal dysplasia in an infant. Both mutant subunits can form functional TRPV6 channels, and the combination of both mutants decreases the calcium uptake. This could be due to the reduced TRPV6 channel expression. Their results also showed that the depleted calcium update might be caused by increased proteases levels, HTRA1 and cathepsin. This study is of great significance for elucidating its molecular mechanism underlying skeletal dysplasia in one patient. The data and conclusion seem convincing. However, the manuscript is not well-organized and somehow overwhelming to read. Much of redundant information could’ve been removed.
Major concerns:
1 Line 95, What exactly is protein dose?
2 Is it possible to make a structure cartoon to show the potential interactions between G660 and W85, between G66 and QQKR83 motif?
3 Line 212, does the dysfunction of the channel result in the altered protein expression or the other way around? As far as I am concerned, for example, it is the reduced expression level that leads to the decreased channel activity or currents, which makes the channel not functional.
4 Line 114, the read-through phenomenon is not necessary to be mentioned here if the authors didn’t plan to talk more about it.
5 Please try to use more short paragraphs and get to the points instead of using one long overwhelming paragraph. This applies to results and discussion sections.
6 Line 107, I thought R510stop mutant is not expressed within the membrane and thus can’t uptake Ca2+ ions?
7 Is it possible to show electrophysiological recordings, for example, currents, from the mutants?
Minor concerns:
1 Line 153, motive should be motif.
2 Line 101 and 108, I am wondering if it’s okay to say (not shown), please check with the guideline for authors.
3 Line 200, maybe put the whole paragraph to the discussion section?
5 Fig 1D, 2A, 2B…, Please color the curves as in Fig 1b to better distinguish them.
6 Line 376, Please provide the reference.
7 Fig 1, could the authors make the figure legend big enough to read? For example, the G660R in 1a, Fig 2,
Author Response
Dear reviewer, we thank for reading the manuscript and constructive criticism. Our step by step reply is given below.
Fecher-Trost et al. here investigated the role of two mutations in TRPV6 channels in the development of skeletal dysplasia in an infant. Both mutant subunits can form functional TRPV6 channels, and the combination of both mutants decreases the calcium uptake. This could be due to the reduced TRPV6 channel expression. Their results also showed that the depleted calcium update might be caused by increased proteases levels, HTRA1 and cathepsin. This study is of great significance for elucidating its molecular mechanism underlying skeletal dysplasia in one patient. The data and conclusion seem convincing. However, the manuscript is not well-organized and somehow overwhelming to read. Much of redundant information could’ve been removed.
Major concerns:
1 Line 95, What exactly is protein dose?
We changed the term to “amount of functional channels”.
2 Is it possible to make a structure cartoon to show the potential interactions between G660 and W85, between G66 and QQKR83 motif?
We added an additional supplementary Fig.1C and mentioned the figure in the text. The figure shows the crystal structure of TRPV6 including the exact position of G660, W85 and K82 of TRPV6.
3 Line 212, does the dysfunction of the channel result in the altered protein expression or the other way around? As far as I am concerned, for example, it is the reduced expression level that leads to the decreased channel activity or currents, which makes the channel not functional.
In fact, this question cannot be answered at the moment. The referee is correct: It is not clear if the dysfunctional channel induces upregulation of certain proteases or if the miss-folding of the channel leads to protease dependent digestion of the TRPV6 channel in the plasma membrane. To solve this needs further investigation.
4 Line 114, the read-through phenomenon is not necessary to be mentioned here if the authors didn’t plan to talk more about it.
If read-through had occurred, the experiments would have had to be interpreted differently. Read-through is a very rare phenomenon that is the reason we wanted to explain it and introduced the citation. We believe the explanation is necessary to understand the logic of the experiment.
5 Please try to use more short paragraphs and get to the points instead of using one long overwhelming paragraph. This applies to results and discussion sections.
We introduced several headlines into results and discussion section.
6 Line 107, I thought R510stop mutant is not expressed within the membrane and thus can’t uptake Ca2+ ions?
The short mutant is expressed and is also located in the plasma membrane (Fig. 1, supplementary Fig. 2A). However, the protein is truncated within the transmembrane domains so that no functional channels with a calcium-conducting pore can be formed.
7 Is it possible to show electrophysiological recordings, for example, currents, from the mutants?
At present we are not able to show representative electrophysiological data of the mutants expressed alone or in combinations. This is an interesting issue being planned to do next. We want to find out if the mutations of the affected child alter the Ca-selectivity of TRPV6.
Minor concerns:
1 Line 153, motive should be motif.
We changed that.
2 Line 101 and 108, I am wondering if it’s okay to say (not shown), please check with the guideline for authors.
The referee is correct. We deleted one sentence which said that the fluophores used (GFP and RFP) did not influence the Ca-measurement. Since this is known it is not necessary to explain or to show additional data. The second term “(not shown)” was simply deleted because we introduced traces showing empty vector transfected cells in figures 1D and supplementary Fig.1A.
3 Line 200, maybe put the whole paragraph to the discussion section?
We have moved this part into the discussion section.
5 Fig 1D, 2A, 2B…, Please color the curves as in Fig 1b to better distinguish them.
We used only colors for monomeric expression (as in Fig.1B) and decided to mark traces derived from co-expression experiments with numbers. We also tried to work with several colored traces but decided that the figures become more confusing. We decided to use split-colored columns to make it easier to understand which constructs were used in the co-expression experiments. At least also from the value of the columns it becomes clear to which trace they belong.
6 Line 376, Please provide the reference.
Done.
7 Fig 1, could the authors make the figure legend big enough to read? For example, the G660R in 1a, Fig 2,
Done.
Reviewer 2 Report
I really appreciate that Fecher-Trost et al. have taken majority of my previous comments into consideration during multiple rounds of revisions of their manuscript. The manuscript has greatly improved. I thank the authors for providing new data using CoIPs and Mass Spectrometry. I do not have any more hesitations in recommending their current work for a publication. Thank you.
Through lines 174 to 184 please ensure that the figure numbers are correct.
Author Response
Dear reviewer, we thank for reading again the manuscript and the very positive feedback. Our reply is given below.
I really appreciate that Fecher-Trost et al. have taken majority of my previous comments into consideration during multiple rounds of revisions of their manuscript. The manuscript has greatly improved. I thank the authors for providing new data using CoIPs and Mass Spectrometry. I do not have any more hesitations in recommending their current work for a publication. Thank you.
Through lines 174 to 184 please ensure that the figure numbers are correct.
Thank you for pointing this out, we have corrected the numbering.
We thank the referee for his intensive work on our manuscript-the comments helped a lot and all the suggestions turned out to be correct.
Submission Date
05 November 2021
Date of this review
16 Nov 2021 13:15:20
Round 2
Reviewer 1 Report
The authors have addressed all my concerns.
This manuscript is a resubmission of an earlier submission. The following is a list of the peer review reports and author responses from that submission.
Round 1
Reviewer 1 Report
Authors examined the role of two TRPV6 mutations in calcium uptake in HEK293 cell line as well as effect of combination of the TRPV6 subunits on calcium uptake in the same cell line. They also reported that expression of proteases increased in the affected placenta and BeWo cells cultured in low calcium concentration. A few concerns are listed below:
- Why HEK293 cell line, not trophoblast cell lines including BeWo, was used to study effect of mutations in the TRPV6 on calcium uptake?
- TRPV6 is mainly expressed in human syncytiotrophoblast. Why were syncytialization of BeWo cells not experimentally induced in this study?
- Why immunohistochemistry did not performed for TRPV6, HTRA1, and cathepsin G using the paraffin-embedded placental sections?
- Expression of proteases HTRA1 and cathepsin G increased in both the affected placenta and BeWo cells cultured in low calcium concentration. Authors claim that proteases act as a compensation mechanism to increase calcium uptake in the placenta with mutations of TRPV6 by affecting intercellular gaps and thus increasing the permeability for calcium ions. However, this reviewer could not understand what the intercellular gap is meaningful in syncytiotrophoblast.
- Proteomics using proteins extracted from paraffin-embedded placental tissue sections is questionable. Was 1.3 fold change considered to be significant? It seems too low.
In the results section, results were described mistakenly in wrong figures. For examples, in line 90, Figure 1B needs to be corrected into Figure 1D. In line 105, Figure 1C needs to be corrected into Figure 1E.
Line 322. The terminology of labyrinth is for mouse placenta not human.
Author Response
Dear reviewer,
we thank you for your constructive critism and added additional experimental data, reorganized figures and made changes in the manuscript text. All changes in the manuscript are highlighted in blue.
please see attached pdf. document

Reviewer 2 Report
Comments on MS 1279167
In this MS, the authors described 2 types of TRPV6 mutation, impairing the assembly of the channel and its function.
Despite the fact that the MS is well written, it suffers from some defaults I describe below, rendering impossible its acceptation at that time without modifications.
My main concern is how the Ca2+ recordings were made: the work is based on the expression of different versions of TRPV6, wt and 2 mutations. To see the Ca2+ uptake, the cells are maintained in a Ca2+-free solution. But everyone in the Ca2+ field knows that the Ca2+-free medium induces a passive Ca2+ store depletion, and next the opening of store-operated Calcium channel. HEK293 cells have a noticeable SOCE. It means that with this protocol, the Ca2+ uptake is due to TRPV6 but also to the SOCE. To complete, I would like to see the same kind of experiments done on wt HEK293 cells, or even better on HEK293 cells transfected with the empty vector. I am pretty sure that 2.5 mM Ca2+ adding on the cells will induce a Ca2+ uptake. Unfortunately, it is not shown as mentioned line 108. It could be interesting for the reader to see also the Ca2+ uptake with HEK293 cells expressing only the R510Stop mutant… due only to the SOCE ? I am also surprised that HEK293 do not detached from the support in Ca2+-free conditions.
The figures and their legends should be reviewed because there are several errors when they are describes in the text. It happens at the first figure. Lines 79 to 81 “Fig 1A… it is Fig 1B ! Furthermore, the figures are really small as they occupied half the width of the pages, rendering them sometimes difficult to analyse. There is also a mistake between fig 2E and F.
Minor concerns:
Introduction: it could be interesting to write in this part with allele mutation comes from the mother, and which one comes from the mother. Furthermore, line 53, “both TRPV6 genes of the infant showed mutations”: it is confusing because there is only one TRPV6 gene, and 2 alleles.
Fig1: sub-fig A is interesting, but I think that a drawing of the channel in the membrane will be more interesting for the reader to place the mutations and to understand their consequences.
Lines 81 to 84 and Fig 1B and C : “Surprisingly…there is a small difference if one compares the AUC (NOT SHOWN)”: it means that the authors made the choice to show a histogram with peak values, that are not different, and not to show an histogram with AUC that could show a difference ??? In this case, both peak values and AUC must be shown.
Lines 86 to 90: “we mimicked the …affected child (fig 1B)”: this is not fig 1B, but 1D. In the next sentence (lines 92-93), the authors wrote that with the 2 mutations, the Ca2+ uptake is reduced by “about 50%”.. not at all ! I would like the authors to give real values and not something “around”. A close view on fig 1 E shows that the peak is decreased from 1.5 to 1.1… far from 50%. Furthermore, like previously, there is no histogram about AUC, so it is impossible for the reader to know exactly what the decrease is due to the mutations… please, give values and show it like for fig 1 C.
Line 147: “it can be concluded”… sure ?
Line 226 and M&M lines 177 to 490: “high (1 mM) or low (0.7 mM)”… the difference between high and low is really tiny… Furthermore, cells were cultures in F-12 Nut medium, and after 24 the medium was changed for one with 0.7 or 1 mM calcium for further 48h. How these Ca2+ concentrations were calculated ? F12 medium contains Ca2+, FBS contains Ca2+…
Author Response

(The authors gave the same response as above.)

Reviewer 3 Report
Please see the attached review report.

Author Response

(The authors gave the same response as above.)

Round 2
Reviewer 1 Report
“We used an algorithm in the Scaffold software programme where the p>0.05 significance threshold is reached at a 1.3 fold protein change. Of course we would take only proteins into account which are more than >2 fold are upregulated in one group. This is the case for cathepsin G and HTRA1, as they were detected exclusively in the affected child during analysis. To the best of our knowledge, there is no work to date that shows a method by which anyone identified as many proteins with mass spectrometry from only a few formalin fixed (3 μm) tissue sections as we have done.”
Comments: The effect of calcium on HTRA4 expression in BeWo cells needs to be validated using other methods such as Western blot, real-time PCR and/or immunocytochemistry.
“BeWo cells endogenously express TRPV6 transcripts but HEK293 cells do not. We did not want to have a TRPV6 background in our transfected cells. In addition BeWo cells are derived from a chorion carcinoma and we know that TRPV6 expressing cells which are derived from a carcinoma express a lot of TRPV6 splice variants. This is the case in the breast cancer cell line, T47D, as well as in cells derived from prostate cancer (unpublished data).”
“We fully agree with the referee and do not think that physical gaps are created in the syncytiotrophoblast layer. But it is conceivable that more extracellular matrix proteins are degraded in the placenta of the affected child, possibly also at the syncytiotrophoblasts or at their contact sites to e.g. cytotrophoblasts. However, we have too little material from the child placenta available for a deeper proteomic analysis or functional protease assays as we have investigated for mouse trophoblasts and placenta (Winter et al. 2020, IJMS).”
Comments: There is lack of physiological relevance of TRPV6 function in human placenta. Only evidence provided for physiological relevance in placenta is effect of calcium, not TRPV6, in BeWo cells on protease expression. No direct evidence demonstrating the role of TRPV6 in BeWo cells was provided, which significantly weakens authors’ main argument.
Although I believe that using cell lines is not an ideal method to recapitulate physiological events and HEK293 cells were utilized to study effect of TRPV6 mutations in this manuscript, at least human placental cell lines should have used to link the role of TRPV6 to the placenta.
I believe that evidence showing localization of TRPV6 protein in the syncytialized BeWo cells as well as effect of TRPV6 on calcium influx and the HTRA4 expression in BeWo cells needs to be provided to improve physiological relevance of TRPV6 function in human placenta.
Reviewer 3 Report
I appreciate the changes made in the manuscript by the authors as per the suggestion of the reviewer. Also, thank you for providing the calcium traces for the empty vector transfected cells.
My biggest issue is that the authors repeatedly mention hetero-tetramers/multimers without providing a direct evidence of such channels at the cell surface. This is a major concern esp. between WT and R510/470stop. I understand that it is tempting to propose this in support of the functional data obtained by co-over-expression of WT with R510stop (paternal) or with G660R (maternal) TRPV6. However, it is very difficult to envision a functional heteromeric TRPV6 channel formed by WT and R510stop. In addition to the points I mentioned earlier, two subunits of such a channel would be missing I575 residues and side chains of four I575 residues, one from each subunit, are needed to form the narrow constriction at the gate region of TRPV6.
The authors in their response letter suggest that supplemental figure 2B shows co-immunoprecipitation of TRPV6 WT and R510stop mutant. However, this is misleading because supplemental figure 2B only suggests that TRPV6510stop makes it to the plasma membrane but notably in a much lower amount compared to WT. However, figure 1 (right) shows an opposite trend with R510stop more expressed than WT. In your favor, the message that the presence of R510stop does not affect the TRPV6 mediated calcium influx remains uncontested. But the evidence that they form heteromeric and functional complexes is rather weak and should be toned down throughout the manuscript.
One likely scenario that could explain authors’ findings could be that R510stop specifically prevents the cell surface trafficking of G660R but not G660A or WT. Therefore in conditions mimicking the child’s combination of G660R + R510stop TRPV6, it is important to expand figure Supplemental 2 by showing the biotinylated G660R and R510stop PM fractions from cells co-expressing these mutants. I insist the authors to provide this data.
On page 3, in first sentence, where the authors state that “It can be seen that …”. Before that the authors should specify that “The parental combinations resulted in comparable Ca2+ fluxes with no significant difference in paternal vs. maternal combinations”
After rephrasing of their earlier sentence in the discussion section, the authors now state that “The maternal combination, TRPV6 WT allele and G660R mutation, does have a minor influence on the function of the channel.” Are the authors comparing the quantified data in figure 1C to 1E? Are the transfected DNA amounts of single TRPV6 construct comparable to co-expressed constructs?